# Renal Toxicities in Cancer Patients Receiving Immune-Checkpoint Inhibitors: A Meta-Analysis

**DOI:** 10.3390/jcm11154373

**Published:** 2022-07-27

**Authors:** Matteo Righini, Veronica Mollica, Alessandro Rizzo, Gaetano La Manna, Francesco Massari

**Affiliations:** 1Nephrology and Dialysis Unit, Santa Maria delle Croci Hospital, AUSL Romagna, 48121 Ravenna, Italy; matteo.righini5@unibo.it; 2Nephrology, Dialysis and Transplantation Unit, IRCCS Azienda Ospedaliero-Universitaria di Bologna, University of Bologna, 40126 Bologna, Italy; gaetano.lamanna@unibo.it; 3Medical Oncology, IRCCS Azienda Ospedaliero-Universitaria di Bologna, Via Albertoni-15, 40138 Bologna, Italy; veronica.mollica7@gmail.com; 4Department of Experimental, Diagnostic and Specialty Medicine, University of Bologna, 40126 Bologna, Italy; 5Struttura Semplice Dipartimentale di Oncologia Medica per la Presa in Carico Globale del Paziente Oncologico “Don Tonino Bello”, Istituto di Ricerca e Cura a Carattere Scientifico (IRCCS), Istituto Tumori Giovanni Paolo II-Bari, 70124 Bari, Italy; rizzo.alessandro179@gmail.com

**Keywords:** immune-checkpoint inhibitors, meta-analysis, renal toxicity, PD-1 inhibitors, PD-L1 inhibitors, chronic kidney disease, acute kidney injury

## Abstract

Aim: We performed a meta-analysis of the available clinical trials of immune-checkpoint inhibitors to assess risk differences and relative risks of renal toxicity. Methods: 17 randomized phase III studies were selected, including 10,252 patients. Results: The administration of immune-checkpoint inhibitors resulted in an overall low-grade, high-grade and all-grade renal toxicity Risk Difference of: 0.746% (95% CI 0.629% to 1.15%, *p* < 0.001—random), 0.61% (95% CI, 0.292–0.929%, *p* < 0.001—fixed) and 1.2% (95% CI, 0.601–1.85%—random), respectively. The pooled Relative Risk of low-grade, high-grade and all-grade renal toxicity was: 2.185 (95% CI 1.515–3.152—fixed), 2.610 (95% CI, 1.409–4.833, *p* = 0.002—fixed) and 2.473 (95% CI, 1.782–3.431, *p* < 0.001—fixed), respectively. An increased risk of renal toxicity was evident in some subgroups more than others. Conclusion: Immune-checkpoint inhibitors are associated with an increased risk of renal toxicity.

## 1. Introduction

In the recent past, the introduction of the immune-checkpoint inhibitors (ICIs) within our anticancer armamentarium has changed, for the better; the natural history of a number of malignancies, greatly improving both prognosis, as well as the overall clinical outcome (including quality of life) of many cancer patients.

To date, we are exploiting for therapeutic purposes, just two immune checkpoints negatively regulating the immune response: the programmed death receptor 1/programmed death receptor ligand 1 (PD-1/PD-L1) axis and the cytotoxic T-lymphocyte antigen 4 (CTLA-4)/CD80 axis.

Indeed, presently available ICIs target these receptor/ligand axes, which are located on the surface of either tumour or immune cells, ultimately leading to a de-inhibition of the immune response which is ontologically devoted to protect our organism from any external harm. As the immune system (and not the tumour itself) is the final “target” of these agents, their toxicity profile is mainly due to its inappropriate activation, which can cross-react against antigens expressed on normal tissues and organs.

Although the overall toxicity profile of ICIs is generally safer, as compared to other anti-cancer agents (such as cytotoxic chemotherapy or targeted agents), and most toxicities are usually mild in severity, still a minority of patients treated with these agents develop life-threatening adverse events, which sometimes are also ill-defined and difficult to recognize [1,2,3,4,5,6,7,8,9].

Renal toxicities are common with both cytotoxic (e.g., cisplatin-induced acute tubular necrosis, leading to acute and chronic kidney failure), and targeted agents (e.g., hypertension and proteinuria from VEGF/VEGFRs-targeting agents). However, the fact that a tight cooperation between oncologists and nephrologist has emerged only in the past few years (with the development of the new subspecialty of Onco-Nephrology [10]), and that, also for the above reason, renal toxicity for years has been defined, within oncological clinical trials, almost only as creatinine increase (which is not a diagnosis, but just a sign), has rendered renal toxicity from ICIs, a “daughter of a lesser God”.

The aim of the present meta-analysis is to investigate if ICIs could raise the risk of renal toxicity in patients with solid tumors, and if we have improved in defining renal toxicities in the modern era of immunotherapy.

## 2. Materials and Methods

### 2.1. Study Selection

The Preferred Reporting Items for Systematic Reviews and Meta-Analyses (PRISMA) statement [11] was followed to identify eligible studies. Two authors (FM, VM) reviewed the citations from PubMed from January 2008 to January 2019 obtained by combining the following words: “Nivolumab” (Bristol-Myers Squibb Pharma EEIG, Dublin, Ireland), “Durvalumab” (AstraZeneca AB, Södertälje, Sweden), “Ipilimumab” (Bristol-Myers Squibb Pharma EEIG, Dublin, Ireland), “Tremelimumab” (AstraZeneca AB, Södertälje, Sweden), “Pembrolizumab” (Merck Sharp & Dohme B.V., Haarlem, Netherland), “Atezolizumab” (Roche Registration GmbH, Grenzach-Wyhlen, Germania) or “Avelumab” (Merck Europe B.V., Amsterdam, Netherland). Our search was limited to human randomized phase III trials published in English focalized on the efficacy and safety of ICIs in cancer patients, which reported data on renal toxicity. Renal toxicity was considered according to RIFLE or AKIN criteria, therefore a primary alteration in serum creatinine >0.3 mg/dl or an increase >150% over the first measurement. In the presence of multiple publications of the same trial, we selected the most recent, or else the most complete in terms of adverse events reporting. Meeting abstracts were not included due to the risk of biases in reporting adverse events.

After checking the safety reports of these studies, renal toxicity was defined as: “creatinine increase”, “renal failure/injury”, or “nephritis”.

### 2.2. Data Extraction, Clinical Outcomes, and Quality Assessment

Three authors (F.M., V.M., A.R.) in an independent manner found and registered data on: authors, study population, experimental and control arms, number and severity of renal adverse events (low- and high-grade).

All trials used either version two, three, or four of the National Cancer Institute’s Common Terminology Criteria for Adverse Events (CTCAE) to report the nature and the severity of the adverse events of interest. The quality of the selected studies was classified according to the Jadad 7-items system based on the randomization process, blinding procedure, and patients’ withdrawal [12].

The primary outcomes of our meta-analysis were: risk differences and relative risks among all trials included, and between specific populations including: patients who received PD-1/PD-L1 or CTLA-4 inhibitors alone, patients who received ICIs in combination with other agents (either chemotherapy or targeted agents), and patients who received CTLA-4 inhibitors alone or in combination with other agents, either immunologic or not.

### 2.3. Statistical Methods

The number of patients receiving ICIs, as well as the number of renal adverse toxicity patients in both treatment and control arms were extracted from all selected studies. Risk Differences (RD) were evaluated as different incidences of renal toxicities between experimental and control arms. Furthermore, we evaluated Relative Risk (RR) as the ratio between incidence of renal toxicity in experimental and control arms; 95% Confidence Intervals (CIs) were subsequently calculated, as proposed by Altmann et al. [13]. Studies with no Immune-mediated adverse events in the treatment or control arms were corrected according to Yates [14].

Cochran’s *Q* statistic was employed to test heterogeneity between studies. The *I*^2 statistic (indicating the percentage of variance in a meta-analysis) was chosen for quantification of inconsistency. Both the inverse variance fixed-effects model (weighted with inverse variance), and the random effect model were adopted to estimate RD and RR among trials included. In case of non-statistically significant heterogeneity, we considered the result obtained by fixed-effects model, while in case of statistically significant heterogeneity results provide by the adoption of the random effect model was considered.

Data collection was obtained using Microsoft Excel (version 16.62, Microsoft, Redmond, Washington, DC, USA), while data analysis was performed with the MedCalc software (MedCalc Software, version 20.113, Ostend, Belgium).

## 3. Results

### 3.1. Search Results

Three authors independently identified a total of 6083 potentially relevant articles on PubMed, EMBASE, Cochrane Library, focused on the efficacy and safety of ICIs in cancer patients. Of these, 5706 were primarily excluded for at least one of the following reasons: review articles, case reports, systematic review, meta-analysis, editorials, letters, or commentaries. Among the 377 remaining studies, only 27 were potentially relevant phase III trials. Of them, 10 were independently excluded by three authors for at least one of the following reasons: presence of ICIs in both experimental and control arm, or inadequate data on the toxicities of interest (Figure 1).

At the end of the process, 17 randomized phase III studies were selected and considered independently by three authors [15,16,17,18,19,20,21,22,23,24,25,26,27,28,29,30,31]. In Table 1, we have summarized the main characteristics of the included trials.

Overall, 10,253 patients have been examined in our analysis; 5721 received ICIs alone (*n* = 3450) or in combination (*n* = 2114) with other agents, 4490 received a PD-1 or PD-L1 inhibitor, while 781 received a CTLA4 inhibitor. All grade renal toxicities have been found in 134 patients enrolled in the experimental arms of the selected trials, and 44 in the control arm. High-grade renal toxicity has been observed in 37 patients enrolled in experimental arms and nine in control patients. The study in which the highest renal toxicity rate has been observed was Keynote-045 [22] in which overall renal toxicity occurred in 38 patients enrolled in experimental arm and 22 patients enrolled in comparator arm.

### 3.2. Quality of Studies

The Jadad scoring system was employed to assess the quality of studies and was investigated by three authors independently. Follow-up time was adequate for all studies. Jadad scores are listed for each trial in Table 1; the mean scores were 4 (range, 3 to 5) underlining the fair quality of the selected studies.

### 3.3. Renal Toxicity among All Clinical Trials

Seventeen studies have been selected for low grade and all grade toxicity analysis [15,16,17,18,19,20,21,22,23,24,25,26,27,28,29,30,31]. Analysis of high-grade toxicity was made on 14 clinical trials [15,16,18,19,20,21,22,23,25,26,27,29,30,31] as three studies [17,24,28] did not reported high-grade toxicity in both experimental and comparator arm. 97 and 35 patients reported low-grade renal toxicity while 37 and 9 patients experienced high-grade toxicity in experimental and comparator arm, respectively.

Overall Risk Difference between experimental and comparator arm was statistically significant in low grade, high-grade and all-grade analysis (Table 2). Also, RR analysis demonstrated a significant higher risk to develop renal toxicity for patients who received an ICI (Table 2). Heterogeneity among trials included in analysis was statistically significant in low and all grade RD analysis (I2: 62%, 52.3%, respectively) while no statistically significant heterogeneity has been observed in G3-5 analysis (I2: 0 %) (Table 3). Kidney toxicity at every grade for each category of ICIs (PD-1/P-L1 and CTLA4) are shown in the Appendix A.

Heterogeneity resulted not statistically significant in low-grade, high-grade and all-grade RR analysis (Table 3). Thus, the administration of ICIs resulted in an overall low-grade, high-grade and all-grade renal toxicity RD of: 0.746% (95% CI 0.629% to 1.15%, *p* < 0.001—random), 0.61% (95% CI, 0.292–0.929%, *p* < 0.001—fixed) and 1.2% (95% CI, 0.601–1.85%—random), respectively (Figure 2). The pooled RR of low-grade, high-grade and all-grade renal toxicity was: 2.185 (95% CI 1.515–3.152—fixed), 2.610 (95% CI, 1.409–4.833, *p* = 0.002—fixed) and 2.473 (95% CI, 1.782–3.431, *p* < 0.001—fixed), respectively (Figure 3).

### 3.4. Renal Toxicity in Patients Receiving ICIs Alone

For this outcome, we considered 10 randomized clinical trials for low- and all-grade analysis [20,21,22,23,24,25,26,28,29,30,31]. Eight clinical trials were considered for high-grade toxicity analysis [21,22,23,25,26,29,30,31] as 2 studies did not report high-grade toxicity in both experimental and comparator arm [24,28]. In this subpopulation, 3401 patients received a PD-1, PD-L1, or CTLA-4 inhibitor alone, while 2456 patients were randomized in the comparator arm. Administration of ICIs resulted in a significant higher risk to develop high-grade renal toxicity (Table 2).

Heterogeneity among study resulted statistically significant in low-grade and all-grade RD analysis (I2: 73.13% and 67.04%, respectively). No statistically significant heterogeneity emerged in high-grade RD analysis and low-, high- and all-grade renal toxicity RR analysis (Table 3). In this population, administration of ICIs leads to a pooled low-, high- and all-grade RD of 0.975% (95% CI, 0.245 to 1.7, *p* < 0.001—random), 0.729% (95% CI, 0.256 to 1.2, *p* = 0.002—fixed) and 0.979% (95% CI 0.165 to 1.79, *p* < 0.001—random), respectively (Table 2). Low-, high- and all-grade toxicity RR was 2.007 (95% CI, 1.349 TO 2.985 *p* = 0.001—fixed), 2.698 (95% CI 1.262 to 5.769 *p* = 0.010—fixed) and 2.257 (95% CI, 1.576 to 3.232 *p* < 0.001—fixed), respectively.

### 3.5. Renal Toxicity in Patients Receiving Combinations of ICIs

Seven studies have been evaluated for this outcome [15,16,17,18,19,20,27]; six studies were also selected for high-grade toxicity analysis [15,16,18,19,20,27]. Overall, 2320 received ICIs in combination with other agents. In this subpopulation, RD analysis revealed a significant correlation in patients receiving combinations of ICIs. RR analysis showed a significant association between immune-checkpoint combination administration and development of low-grade and all-grade toxicity. No statistically significant RR has been found in high-grade analysis (Table 2). Heterogeneity between studies included resulted statistically significant only in all grade RD analysis (I2: 70.9%). In this population, low-, high- and all-grade RD was: 0.627% (95% CI, 0.246 to 1.01 *p* = 0.001—fixed), 0.459% (0.0563 to 0.862, *p* = 0.025—fixed) and 0.979% (95% CI, 0.165% to 1.79—random). RR analysis showed a pooled RR of 3.318 (95% CI, 1.265 to 8.702 *p* = 0.015—fixed) for low-grade toxicity and a pooled RR of 3.576 (95% CI, 1.605 to 7.9689, *p* = 0.002—fixed) in all-grade toxicity (Table 2).

### 3.6. Renal Toxicity in Patients Receiving Anti CTLA-4 Agents

Only two trials were considered for this outcome [20,31], for a total of 781 patients. In this subpopulation RD analysis showed a statistically significant risk difference in low-grade analysis while no statistically significant risk difference was found in high-grade and all-grade analyses. RR analysis revealed a statistically significant RR only in all-grade toxicity (Table 2). Heterogeneity was significant only in all-grade RR analysis (Table 3). Low grade RD was 0.74% (95% CI, 0.15–1.34 *p* = 0.014—fixed). Pooled RR in all-grade toxicity was: 2.137 (95% CI 1.087 to 4.199 *p* = 0.028—random)

## 4. Discussion

We carried out a meta-analysis of 17 randomized clinical trials to evaluate the incidence of renal toxicity among patients receiving ICIs.

In our analysis, we demonstrated that the administration of ICIs is associated to an increased risk of renal toxicity of low- and high-grade. Moreover, this correlation was confirmed also in a specific subpopulation of patients including patients who received ICIs alone and in patients receiving ICIs in combination with other agents. Regarding studies exploring CTLA-4 inhibitors, no statistical significant differences in terms of RD (for high- and all-grade toxicity) and RR (for low- and high-grade toxicities) were found. However, the relatively limited number of trials analysed could not have the adequate power to detect a difference between incidences of these side events.

The estimated incidence of ICIs associated AKI (ICIs-AKI) ranges from 1.4% to 4.9%, and the most common renal lesion is represented by tubulointerstitial nephritis (TIN). Nevertheless, several other immune-mediated pathologies have been associated, including various glomerulonephritides [32,33,34]. Though it is difficult to describe a ICIs induced nephropathy because, to our knowledge, none of these medications showed a direct effect on any kidney structure neither biopsy proven studies.

The study conducted by Cortazar et al. shows that there is a variable and often prolonged delay between ICIs initiation and the development of AKI, that the frequent rate of extrarenal immune-related adverse events (irAEs) occurring concomitantly or immediately preceding the AKI, that patients who have concomitant extrarenal irAEs have a lower likelihood of kidney recover, that patients who were rechallenged with ICIs therapy did not develop recurrence of ICIs-AKI and, as expected, that those who fail to achieve kidney recovery after ICIs-AKI show increased mortality [35]. In a large retrospective study conducted by Seethapathy et al., the authors underline that irAEs may be a novel risk factor for adverse kidney outcomes, since those patients resulted at increased risk of sustained eGFR loss. Those findings suggests that adverse kidney outcomes after ICIs may have a different mechanism than typical TIN, however, has to be considered that these patients are not exempted from other causes of AKI such as volume contraction, ischemic tubular injury or obstructive nephropathy [36]. In a recent metanalysis conducted by Magee DE et al., the authors describes the adverse event profile for immunotherapy agents compared with chemotherapy in solid organ tumors. The authors found an incidence of AKI in 1.31% of the studied patients that is lower than ours. Nevertheless the authors choose only the studies that compared treatment with ICI with standard chemotherapeutic regimens; therefore it would be possible that other studies considering ICIs alone or in combination with other agents influenced our results [37]. Therefore, is of utmost importance to perform long term trials to analyze acute and chronic renal toxicity of ICIs.

Our analysis was limited by several factors, including the different setting, disease and modality of administration (combination or alone) of ICIs. It is thus not surprising that higher renal toxicity rates have been found in the Keynote-045 study, which compared pembrolizumab to investigator-choice chemotherapy in patients progressed to standard first line therapy for advanced or metastatic urothelial carcinoma [22]. It is indeed probable that the previous administration of platinum compounds (as well as the specific disease site) could partially justify this finding.

Furthermore, our meta-analysis per definition cannot capture the possible role of a pre-existing subclinical renal damage, due to co-morbidities (e.g., hypertension and diabetes) or previous therapeutic interventions (e.g., the above mentioned renal damage from previous platinum-based therapies or from previous nephrectomy). The estimation of the exact incidence of renal toxicity among patients receiving ICIs is of particular interest, since these agents are often administered in patients with pre-existing renal failure of some degree.

Here, two other key issues emerge.

First of all, very limited data are available about the administration of ICIs in patients with moderate or severe renal failure, or even in patients with end-stage renal disease on dialysis. In urothelial cancer, ICIs have been administered in patients unfit to receive standard platinum-based chemotherapy, thus allowing the enrolment of patients with a certain degree of impaired renal function, ultimately demonstrating that the immunological treatment did not raise the risk of renal failure [38]. Unfortunately, the creatinine or creatinine clearance limits imposed by the vast majority of randomized controlled phase III trials do not allow for the confirmation of the activity and, more importantly, safety of ICIs in patients with more advanced renal impairment. From a pharmacological viewpoint, this makes little sense, if any, considering that ICIs are monoclonal antibodies which are cleared by the reticulo-endothelial system, and not by the kidneys [38]. Indeed, retrospective case series of cancer patients on dialysis treated with these agents are accumulating, suggesting the safety of this therapeutic approach [39,40,41,42].

Another point is that, in trials conducted in the era of modern immunotherapy, renal toxicity is almost always reported as “creatinine increase” that is extremely incomplete.

Not to take into account that “You do not see what you’re not looking for”, meaning that immune-related renal toxicity is, in our opinion, often under-diagnosed. It is indeed common to observe, during the first months of treatment with ICIs, an initial rise in creatinine levels, which subsequently stabilizes, which could indeed be due to an autoimmune response against the kidneys.

## 5. Conclusions

As suggested by our study, ICIs are associated with an increased risk of renal toxicity. Although usually of a low-grade, these adverse events are, in our opinion, too often underdiagnosed. Thus, the risk of renal toxicity should be always considered before treatment starts, and the cooperation of nephrologist and oncologist has to be encouraged in the presence of a deterioration in renal function. For sure, the use of ICIs should not be discouraged in patients with pre-existing renal impairment, even when on dialysis.

### Summary Points

In recent years, immune-checkpoint inhibitors revolutionized the treatment scenario and outcomes of cancer patients. Few studies have exanimated the effective incidence of renal toxicity mediated by immune-checkpoint inhibitors.

We selected clinical trials evaluating safety and efficacy of immune-checkpoint inhibitors in patients with solid tumors. We identified 17 randomized phase III studies, including 10,252 patients. Of these, 5721 received ICIs alone (*n* = 3450) or in combination (*n* = 2114) with other agents, 4490 received a PD-1 or PD-L1 inhibitor, while 781 received a CTLA4 inhibitor.

We performed a meta-analysis aimed at evaluating the Risk Difference (as different incidences of renal toxicities between experimental and control arms) and Relative Risk (as the ratio between incidence of renal toxicity in experimental and control arms).

In our study, the administration of immune-checkpoint inhibitors resulted in an overall low-grade, high-grade and all-grade renal toxicity risk difference of 0.746% (95% CI 0.629% to 1.15%, *p* < 0.001—random), 0.61% (95% CI, 0.292–0.929%, *p* < 0.001—fixed) and 1.2% (95% CI, 0.601–1.85%—random), respectively. The pooled relative risk of low-grade, high-grade and all-grade renal toxicity was 2.185 (95% CI 1.515–3.152—fixed), 2.610 (95% CI, 1.409–4.833, *p* = 0.002—fixed) and 2.473 (95% CI, 1.782–3.431, *p* < 0.001—fixed), respectively.

Administration of ICIs alone leads to a pooled low-, high- and all-grade RD of 0.975% (95% CI, 0.245 to 1.7, *p* < 0.001 –random), 0.729% (95% CI, 0.256 to 1.2, *p* = 0.002—fixed) and 0.979% (95% CI 0.165 to 1.79, *p* < 0.001—random), respectively. Low-, high- and all-grade toxicity RR was 2.007 (95% CI, 1.349 to 2.985 *p* = 0.001—fixed), 2.698 (95% CI 1.262 to 5.769 *p* = 0.010—fixed) and 2.257 (95% CI, 1.576 to 3.232 *p* < 0.001—fixed), respectively.

In patients receiving immune-checkpoint inhibitors in combination with other agents, low-, high- and all-grade RD was 0.627% (95% CI, 0.246 to 1.01 *p* = 0.001—fixed), 0.459% (0.0563 to 0.862, *p* = 0.025—fixed) and 0.979% (95% CI, 0.165% to 1.79—random). RR analysis showed a pooled RR of 3.318 (95% CI, 1.265 to 8.702 *p* = 0.015—fixed) for low-grade toxicity and a pooled RR of 3.576 (95% CI, 1.605 to 7.9689, *p* = 0.002—fixed) in all-grade toxicity.

Immune-checkpoint inhibitors are associated with an increased risk of renal toxicity of low- and high-grade.

An increased risk of renal toxicity was evident in specific subpopulation of patients, such as patients that received immune-checkpoint inhibitors alone or those receiving immune-checkpoint inhibitors in combination with other agents.

## Figures and Tables

**Figure 1 jcm-11-04373-f001:**
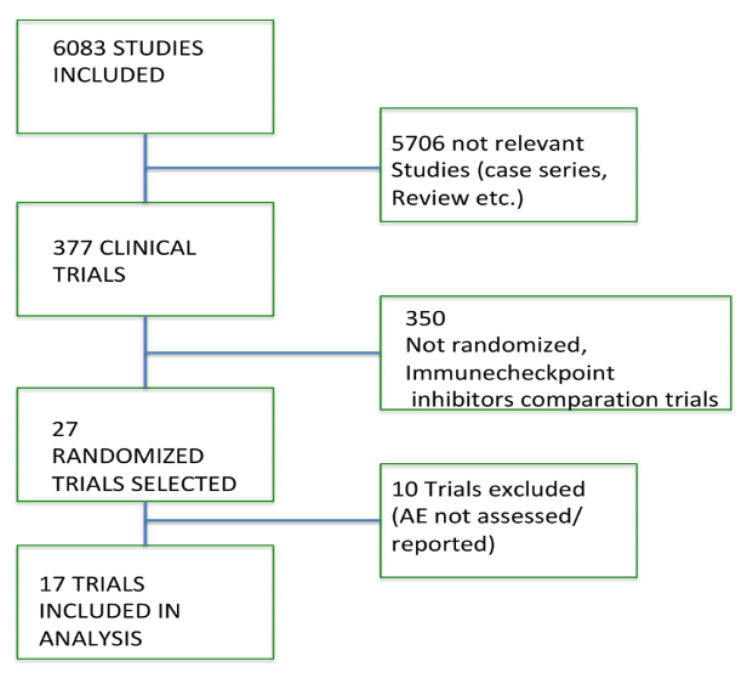
Studies screened and included in our meta-analysis.

**Figure 2 jcm-11-04373-f002:**
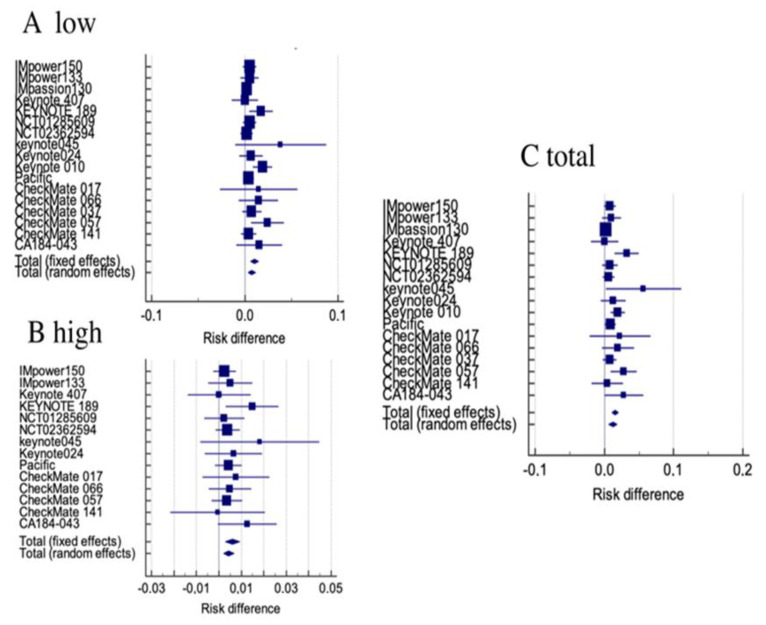
Forest plot of RD among patients receiving immune–checkpoint inhibitors. (**A**) Low grade toxicity, (**B**) high grade toxicity, (**C**) all grade toxicity.

**Figure 3 jcm-11-04373-f003:**
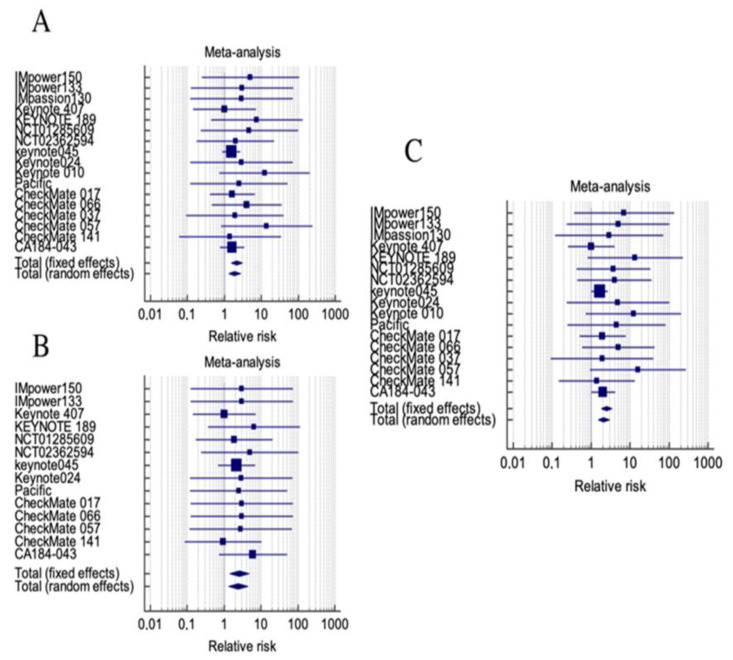
Forest plot of RD among patients receiving immune–checkpoint inhibitors. (**A**) Low grade toxicity, (**B**) high grade toxicity, (**C**) all grade toxicity.

**Table 1 jcm-11-04373-t001:** Studies included; for each study, we reported: study name, phase, experimental and comparator treatment arm, disease, setting, population on study, safety population and JADED score.

Study Selected	Phase	First Name	Disease	Setting	Experimental Arm n/Safety Population	Comparator Arm n/Safety Population	Jaded Score
IMpower150 [14]	III	Socinski MA	Non squamous NSCLC	First line	Atezolizumab + Bevacizumab + Carboplatin + Paclitaxel 800/393	Bevacizumab + Carboplatin + Paclitaxel 400/394	3
IMpower133 [15]	III	Horn L	SCLC	First line	Atezolizumab + Carboplatin + Etoposide 201/198	Carboplatin + Etoposide 202/196	5
IMpassion130 [16]	III	Schmid P	Breast cancer	First line	Atezolizumab + Nab Paclitaxel 452/451	Placebo + Nab paclitaxel 451/438	5
Keynote 407 [17]	III	Paz-Ares L	Squamous NSCLC	First line	Pembrolizumab + Carboplatin + Nab paclitaxel or paclitaxel 278/278 278	Placebo + Carboplatin + Nab paclitaxel or paclitaxel 281/280 281	5
Keynote 189 [18]	III	Gandhi L	non squamous NSCLC	First line	Pembrolizumab + Pemetrexed + Platinum compounds 410/405	Placebo + Pemetrexed + Platinum compounds 206/202	5
NCT01285609 [19]	III	Govindan R	squamous NSCLC	First line	Ipilimumab + Paclitaxel 388/388	Placebo + Paclitaxel 361/361	5
NCT02362594 [20]	III	Eggermont AMN	melanoma	adjuvant	Pembrolizumab 514/509	Placebo 505/502	5
Keynote045 [21]	III	Bellmunt J	urothelial carcinoma	Previously treated	Pembrolizumab 270/266	Docetaxel or Vinflunine or Paclitaxel 272/255	3
Keynote024 [22]	III	Reck M	NSCLC	First line	Pembrolizumab 1,547,154	Platinum based therapy 151/150	3
Keynote 010 [23]	II/III	Herbst RS	NSCLC	Previously treated	Pembrolizumab 690/682	Docetaxel 343/309	3
Pacific [24]	III	Antonia SJ	NSCLC	maintenance	Durvalumab 476/475	Placebo 237/234	5
CheckMate 017 [25]	III	Brahmer J	squamous NSCLC	Previously treated	Nivolumab 135/131	Docetaxel 137/129	3
CheckMate 066 [26]	III	Robert C	melanoma	First line	Nivolumab + Dacarbazine 210/206	Placebo + Dacarbazine 208/205	4
CheckMate 037 [27]	III	Weber JS	melanoma	Previously treated	Nivolumab 272/268	Dacarbazine or Paclitaxel + Carboplatin 133/102	3
CheckMate 057 [28]	III	Borghaei H	non squamous NSCLC	Previously treated	Nivolumab 292/287	Docetaxel 290/268	3
CheckMate 141 [29]	III	Ferris RL	squamous head and neck	Previously treated	Nivolumab 240/236	Docetaxel or methotrexate or Cetuximab 121/111	3
CA184-043 [30]	III	Kwan ED	mCRPC	Previously treated	Ipilimumab 399/393	Placebo 400/396	5

NSCLS: Non small cells lung cancer. SCLS: Small cells lung cancer.

**Table 2 jcm-11-04373-t002:** Summary of the results of our meta-analysis. We reported RD and RR of each study included with 95% CI. We also reported final results estimated adopting fixed and random model effect of overall population, population receiving ICIs alone, ICIs combination, and CTLA-4 inhibitors. Results were estimated for low-grade toxicity, high-grade toxicity and all-grade toxicity.

Study	G1-2 RD	G1-2 RR	G3-5 RD	G3-5 RR	Gtot RD	Gtot RR
IMpower150 [14]	0.00509 (−0.00195 to 0.0121)	5.013 (0.241 to 104.082)	0.00254 (−0.00244 to 0.00753)	3.008 (0.123 to 73.609)	0.00763 (−0.000972 to 0.0162)	7.018 (0.364 to 135.422)
IMpower133 [15]	0.00505 (−0.00482 to 0.0149)	2.970 (0.122 to 72.465)	0.00505 (−0.00482 to 0.0149)	2.970 (0.122 to 72.465)	0.0101 (−0.00383 to 0.0240)	4.950 (0.239 to 102.448)
IMpassion130 [16]	0.00221 (−0.00212 to 0.00654)	2.907 (0.119 to 71.179)	NE	NE	0.00221 (−0.00212 to 0.00654)	2.907 (0.119 to 71.179)
Keynote 407 [17]	0.0000514 (−0.0139 to 0.0141)	1.007 (0.143 to 7.100)	0.0000514 (−0.0139 to 0.0141)	1.007 (0.143 to 7.100)	0.000103 (−0.0196 to 0.0198)	1.007 (0.254 to 3.987)
Keynote 189 [18]	0.0173 (0.00459 to 0.0300)	7.500 (0.430 to 130.674)	0.0148 (0.00305 to 0.0266)	6.500 (0.368 to 114.819)	0.0321 (0.0149 to 0.0493)	13.500 (0.807 to 225.965)
NCT01285609 [19]	0.00515 (−0.00197 to 0.0123)	4.653 (0.224 to 96.597)	0.00238 (−0.00657 to 0.0113)	1.861 (0.169 to 20.435)	0.00754 (−0.00388 to 0.0190)	3.722 (0.418 to 33.143)
NCT02362594 [20]	0.00194 (−0.00475 to 0.00863)	1.972 (0.179 to 21.685)	0.00393 (−0.00151 to 0.00936)	4.931 (0.237 to 102.467)	0.00587 (−0.00274 to 0.0145)	3.945 (0.442 to 35.175)
Keynote045 [21]	0.0384 (−0.0105 to 0.0873)	1.544 (0.880 to 2.711)	0.0181 (−0.00840 to 0.0447)	2.157 (0.673 to 6.917)	0.0566 (0.00221 to 0.111)	1.656 (1.008 to 2.720)
Keynote024 [22]	0.00649 (−0.00619 to 0.0192)	2.923 (0.120 to 71.185)	0.00649 (−0.00619 to 0.0192)	2.923 (0.120 to 71.185)	0.0130 (−0.00489 to 0.0309)	4.871 (0.236 to 100.628)
Keynote 010 [23]	0.0191 (0.00880 to 0.0293)	12.255 (0.731 to 205.500)	NE	NE	0.0191 (0.00880 to 0.0293)	12.255 (0.731 to 205.500)
Pacific [24]	0.00421 (−0.00161 to 0.0100)	2.468 (0.119 to 51.213)	0.00421 (−0.00161 to 0.0100)	2.468 (0.119 to 51.213)	0.00842 (0.000203 to 0.0166)	4.443 (0.240 to 82.188)
CheckMate 017 [25]	0.0149 (−0.0270 to 0.0568)	1.641 (0.400 to 6.726)	0.00763 (−0.00727 to 0.0225)	2.955 (0.121 to 71.868)	0.0225 (−0.0217 to 0.0668)	1.969 (0.503 to 7.708)
CheckMate 066 [26]	0.0145 (−0.00658 to 0.0357)	3.981 (0.449 to 35.312)	0.00485 (−0.00464 to 0.0143)	2.986 (0.122 to 72.866)	0.0194 (−0.00368 to 0.0425)	4.976 (0.586 to 42.221)
CheckMate 037 [27]	0.00746 (−0.00284 to 0.0178)	1.914 (0.0927 to 39.542)	NE	NE	0.00746 (−0.00284 to 0.0178)	1.914 (0.0927 to 39.542)
CheckMate 057 [28]	0.0244 (0.00654 to 0.0422)	14.010 (0.804 to 244.139)	0.00348 (−0.00333 to 0.0103)	2.802 (0.115 to 68.492)	0.0279 (0.00883 to 0.0469)	15.878 (0.921 to 273.783)
CheckMate 141 [29]	0.00424 (−0.00405 to 0.0125)	1.418 (0.0582 to 34.530)	−0.000534 (−0.0216 to 0.0206)	0.941 (0.0862 to 10.265))	0.00370 (−0.0190 to 0.0264)	1.411 (0.148 to 13.413)
CA184-043 [30]	0.0155 (−0.00945 to 0.0404)	1.612 (0.741 to 3.509)	0.0127 (−0.000350 to 0.0258)	6.046 (0.731 to 49.989)	0.0282 (0.000299 to 0.0561)	2.015 (0.991 to 4.100)
Tot fixed	0.0105 (0.00629 to 0.0146) ***p* < 0.001**	2.185 (1.515 to 3.152) ***p* < 0.001**	0.00610 (0.00292 to 0.00929) ***p* < 0.001**	2.610 (1.409 to 4.833) ***p* = 0.002**	0.0153 (0.0105 to 0.0201) ***p* < 0.001**	2.473 (1.782 to 3.431) ***p* < 0.001**
Tot Randomized	0.00746 (0.00337 to 0.0115) ***p* < 0.001**	1.893 (1.303 to 2.751) ***p* = 0.001**	0.00444 (0.00216 to 0.00673) ***p* < 0.001**	2.413 (1.282 to 4.543) ***p* = 0.006**	0.0122 (0.00601 to 0.0185) ***p* < 0.001**	2.107 (1.509 to 2.942) ***p* < 0.001**
**IMMUNECHECKPOINT INHIBITOR ALONE**						
RD Tot fixed	0.0138 (0.00694 to 0.0206) ***p* < 0.001**	2.007 (1.349 to 2.985) ***p* = 0.001**	0.00729 (0.00256 to 0.0120) ***p* = 0.002**	2.698 (1.262 to 5.769) ***p* = 0.010**	0.0195 (0.0119 to 0.0272) ***p* < 0.001**	2.257 (1.576 to 3.232) ***p* < 0.001**
RD Tot Randomized	0.00975 (0.00245 to 0.0170) ***p* = 0.009**	1.760 (1.176 to 2.634) ***p* = 0.006**	0.00481 (0.00173 to 0.00790) ***p* = 0.002**	2.551 (1.170 to 5.561) ***p* = 0.018**	0.0141 (0.00558 to 0.0226) ***p* = 0.001**	1.995 (1.387 to 2.870) ***p* < 0.001**
**COMBINATION**						
RD Tot fixed	0.00627 (0.00246 to 0.0101) ***p* = 0.001**	3.318 (1.265 to 8.702) ***p* = 0.015**	0.00459 (0.000563 to 0.00862) ***p* = 0.025**	2.454 (0.856 to 7.033) *p* = 0.095	0.00992 (0.00496 to 0.0149) ***p* < 0.001**	3.576 (1.605 to 7.9689 ***p* = 0.002**
RD Tot Randomized	0.00501 (0.00129 to 0.00872) ***p* = 0.008**	2.955 (1.091 to 8.002) ***p* = 0.033**	0.00399 (0.000592 to 0.00739) ***p* = 0.021**	2.167 (0.733 to 6.406) *p* = 0.162	0.00979 (0.00165 to 0.0179) ***p* = 0.018**	2.828 (1.216 to 6.579) ***p* = 0.016**
**CTLA-4**						
Tot Fixed	0.00746 (0.00150 to 0.0134) ***p* = 0.014**	1.763 (0.834 to 3.723) *p* = 0.137	0.00907 (−0.00206 to 0.0202) *p* = 0.110	3.912 (0.840 to 18.221) *p* = 0.082	0.0181 (0.00278 to 0.0335) ***p* = 0.021**	2.163 (1.103 to 4.242) ***p* = 0.025**
Tot Randomized	0.00565 (0.0000706 to 0.0112) ***p* = 0.47**	1.721 (0.810 to 3.656) *p* = 0.158	0.00879 (−0.00310 to 0.0207) *p* = 0.147	3.611 (0.740 to 17.613) *p* = 0.112	0.0159 (−0.0130 to 0.0449) *p* = 0.280	2.137 (1.087 to 4.199) ***p* = 0.028**

RD, Risk Difference; RR, Relative Risk; G, Grade; tot, total; CI, Confidence Intervals; ICIs, Immune-Checkpoint Inhibitors; CTLA-4, cytotoxic T-lymphocyte antigen 4, NE, Not examinated.

**Table 3 jcm-11-04373-t003:** Heterogeneity among studies included in our meta-analysis.

Population	Q	DF	Significance Level	I2	95% I2
**RD analysis**					
Overall G1-2	42.9206	16	***p* = 0.0003**	62.72%	37.10 to 77.91
Immunecheckpoint Inhibitor Alone G1-2	33.4909	9	***p* = 0.0001**	73.13%	49.33 to 85.75
Combination G1-2	7.8762	6	*p* = 0.24	23.82%	0.00 to 66.37
CTLA-4 G1-2	3.7443	1	*p* = 0.0530	73.29%	0.00 to 93.98
Overall G 3-5	9.3461	13	*p* = 0.7463	0.00%	0.00 to 37.57
Immunecheckpoint Inhibitors Alone G3-4	5.5952	7	*p* = 0.58	0.00%	0.00 to 59.86
Combination G3-5	4.1982	5	*p* = 0.52	0%	0.00 to 70.65
CTLA-4 G3-5	1.0972	1	*p* = 0.2949	8.86%	8.86 to 8.86
Overall Tox	62.5155	16	***p* < 0.0001**	74.41%	58.81 to 84.10
Immunecheckpoint Inhibitors Alone Overall	27.3042	9	***p* = 0.0012**	67.04%	35.85–83.06%
Combination Tox	20.6177	6	***p* = 0.0021**	70.90%	36.57 to 86.65
CTLA-4 overall Tox	3.8129	1	*p* = 0.0509	73.77%	0.00 to 94.08
**RR analysis**					
Overall G1-2	7.5923	16	*p* = 0.96	0%	0 to 0
Immunecheckpoint Inhibitors Alone G1-2	4.6905	9	*p* = 0.86	0%	0.00 to 28.17
Combination G1-2	1.9013	6	*p* = 0.9286	0%	0.00 to 9.40
CTLA-4 G1-2	0.4440	1	*p* = 0.50	0%	0 to 0
Overall G 3-5	2.9917	13	*p* = 0.99	0%	0 to 0
Immunecheckpoint Inhibitors Alone G3-5	1.6102	7	*p* = 0.97	0%	0 to 0
Combination G3-5	4.4643	6	*p* = 0.61	0%	0.00 to 61.41
CTLA-4 G3-5	0.5325	1	***p* = 0.46**	0%	0.00 to 0.00
Overall Tox	10.8636	16	*p* = 0.81	0%	0.00 to 28.04
Immunecheckpoint Inhibitors Alone Overall	5.7022	9	*p* = 0.76	0%	0.00 to 40.91
Combination Tox	1.3359	5	*p* = 0.93	0%	0.00 to 7.76
CTLA-4 overall Tox	0.2747	1	*p* = 0.60	0%	0.00 to 0.00

RD, Risk Difference; RR, Relative Risk; G, Grade; tox, toxicity; CI, Confidence Intervals; ICIs, Immune-Checkpoint Inhibitors; CTLA-4, cytotoxic T-lymphocyte antigen 4.

## Data Availability

Not applicable.

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
