# Peer review of "Renal Toxicities in Cancer Patients Receiving Immune-Checkpoint Inhibitors: A Meta-Analysis"

_jcm, 2022, doi:10.3390/jcm11154373_

Round 1
Reviewer 1 Report
The authors conducted a meta-analysis study regarding the renal toxicities in cancer patients receiving immune-checkpoint inhibitors. The manuscript is well written and the results well presented. I have some comments which may improve the quality of the study:
1. How renal toxicity and the grade of renal toxicity was defined in each study? Please explain in the manuscript.
2. Why the authors did not check the effects of Cemiplimab as an immune-checkpoint inhibitor?
3. Could you please provide a Table with th analyzed Jaded score for each study, including the score in each Jadad itme?
Author Response
- Thank your very much for your question, we added in red our criterias. Renal toxicity was considered according to RIFLE or AKIN criterias, therefore a primary alteration in serum creatinine>0.3 mg/dl or an increase > 150% over the first measurement.
- Thank you for this question. No data on the toxicity of cemiplimab were available at the time of the research, and thus, cemiplimab was not included.
- Dear Reviewer, thank you for this question regarding the Jadad score. We choose to include the aggregate Jadad since all the included trials were phase II/III and phase III clinical trials. In addition, since the Jadad scale has important limitations to be considered, we discussed this point in the Discussion section, as also suggested by the other Reviewer.
Reviewer 2 Report
The authors investigated the risk of renal toxicity in cancer patients receiving ICIs by systematic review procedure. They included 17 RCTs, and performed statistical analyses and concluded that use of ICIs was significantly associated with renal toxicity. However, several important problems are detected in this study. In particular, essential steps of systematic review and meta-analyses should be re-checked at first.
1. Is this study a systematic review? or meta-analysis? If this is a systematic review, did authors register the protocol of this study to PROSPERO or somewhere?
2. Why did authors search the RCTs only by PubMed? I think that the authors should search for the applicable RCTs in Embase, Medline, Cochran etc., too.
3. What is the most important issue that the authors want to clarify? The purpose of this study is obscure. In addition, according to the purpose of the study, the selection criteria of RCTs may be different. For example, if the authors want to evaluate the risk of renal toxicity of ICIs comparing to cytotoxic chemotherapy, the authors should exclude the RCTs whose control arm is placebo-only. This study cannot evaluate the association between use of ICIs and renal toxicity because the study design and RCTs that were evaluated in this study are not appropriate. The authors should consider the followings before evaluate renal toxicity and select the appropriate RCTs. (a) Cancer itself can cause kidney dysfunction due to urinary tract obstruction, tumor lysis syndrome or cancer induced cachexia or dehydration. (b) Cisplatin and VEGF inhibitor can cause renal toxicity.
4. The evaluation of risk of bias is not enough. Jadad system does not consider the concealment. The authors should use GRADE approach or something. In addition, these assessments should be done by at least two persons, independently.
5. Similar investigation has already been published (Magge et al. PMID: 31912796) and their results were different from current article. Therefore, the authors should cite this research and discuss why the results were different.
Author Response
- Thank you for this question. The current paper is a meta-analysis, as reported (red).
- Thank you for catching this oversight. In effect, we made a research on three different databases (red), as you could find in the revised paper.
- The purpose of the study is to investigate if ICIs could raise the risk of renal toxicity in patients with solid tumors, and if we have improved in defining renal toxicities in the modern era of immunotherapy.
We decided to perform a large meta-analysis to collect several studies and to understand whether ICIs could determine renal toxicity alone, in combination with other ICIs or in combination with other agents, regardless of the pathological mechanism that lead there.
We considered that cancer could cause itself kidney disfunction through several mechanisms and that cisplatin and VEGF inhibitor can cause renal toxicity, nevertheless these are complications that could be faced with every pharmacological agent. It would have been impossible to split off ICIs related adverse events that determine renal toxicity only through an immunological mechanism (tubule-interstitial toxicity or acute tubular necrosis) because the only way to do so was through a pathological analysis that wasn’t available for all of these studies.
Considering renal toxicity as a primary alteration in serum creatinine > 0.3 mg/dl or an increase >150% over the first measurement, we choose to perform this analysis.
- Thank you for this comment. We better specified that the Jadad assessment was performed independently by three authors. At the same time, we are aware Jadad may have some limitations and it is sometimes considered quite simplistic and maybe places too much emphasis on blinding. In addition, Jadad does not take into account allocation concealment, which represents a key point to consider in order to avoid bias. At the same time, the Jaded score is widely used to evaluate the general quality of trials and allow to set a minimum standard for the paper’s results to be included in a quantitative analysis such as a meta-analysis. Thank you for your comprehension.
- Thank you very much for the suggestion, we have considered this article in our research and we think it improved our paper. In a recent metanalysis conducted by Magee DE et al. the authors describes the adverse event profile for immunotherapy agents compared with chemotherapy in solid organ tumors. The authors found an incidence of AKI in 1.31% of the studied patients that is lower than ours. Nevertheless the authors choose only the studies that compared treatment with ICI with standard chemotherapeutic regimens, therefore it would be possible that other studies considering ICIs alone or in combination with other agents influenced our results.
Round 2
Reviewer 1 Report
The authors have adequately reported to my comments. I have no further comments.
Reviewer 2 Report
I understood that their article is not systematic review but meta-analysis.
In that case, authors have addressed my queries and improved their article.
I have no additional comments.